# Conformational Analysis of Charged Homo-Polypeptides

**DOI:** 10.3390/biom13020363

**Published:** 2023-02-15

**Authors:** Lavi S. Bigman, Yaakov Levy

**Affiliations:** Department of Chemical and Structural Biology, Weizmann Institute of Science, Rehovot 7610001, Israel

**Keywords:** intrinsically disordered proteins, polyelectrolytes, D/E repeats, K/R repeats, molecular dynamics simulations

## Abstract

Many proteins have intrinsically disordered regions (IDRs), which are often characterized by a high fraction of charged residues with polyampholytic (i.e., mixed charge) or polyelectrolytic (i.e., uniform charge) characteristics. Polyelectrolytic IDRs include consecutive positively charged Lys or Arg residues (K/R repeats) or consecutive negatively charged Asp or Glu residues (D/E repeats). In previous research, D/E repeats were found to be about five times longer than K/R repeats and to be much more common in eukaryotes. Within these repeats, a preference is often observed for E over D and for K over R. To understand the greater prevalence of D/E over K/R repeats and the higher abundance of E and K, we simulated the conformational ensemble of charged homo-polypeptides (polyK, polyR, polyD, and polyE) using molecular dynamics simulations. The conformational preferences and dynamics of these polyelectrolytic polypeptides change with changes in salt concentration. In particular, polyD and polyE are more sensitive to salt than polyK and polyR, as polyD and polyE tend to adsorb more divalent cations, which leads to their having more compact conformations. We conclude with a discussion of biophysical explanations for the relative abundance of charged amino acids and particularly for the greater abundance of D/E repeats over K/R repeats.

## 1. Introduction

The intrinsically disordered regions (IDRs) of proteins are linked to various biological functions [1,2,3,4] and are often characterized by highly charged amino acid content. The more highly charged content of IDRs compared with foldable sequences favors interactions with the solvent and may disfavor their folding into a unique three-dimensional structure [5,6,7,8]. The structural and dynamic properties of IDRs depend on their charge composition. IDRs differ from each other with respect to the fraction of positively and negatively charged residues they contain, their overall net charges, and the organization or pattern of charges along the IDRs. Charge composition and organization are expected to determine the biophysical characteristics and function of IDRs. For example, it was shown that changing charge organization in the IDRs of DNA-binding proteins can tune binding affinity to DNA and the diffusion coefficient for linear diffusion along DNA [9,10,11,12]. In another example, the charge pattern was shown to have a pronounced effect on the ability of IDRs to form condensates via liquid–liquid phase separation [13,14] and on the stability of a complex formed between two highly but oppositely charged intrinsically disordered proteins (IDPs) [15].

IDRs are found to span a wide range of net charges, with the net charge per residue ranging between −1 and +1. For many IDRs, the net charge per residue is close to zero, reflecting the presence of a similar number of negatively and positively charged residues (i.e., polyampholytic IDRs). Other IDRs are highly charged, and their net charge per residue deviates from zero. It was reported that highly negatively charged IDRs are longer and more highly charged than positively charged IDRs [16]. A particularly interesting group of IDRs are those with net charge close to −1 or +1. In these cases, the fraction of negatively or positively charged residues is close to unity. These IDRs, which are quite homogenously charged and are thus classified as polyelectrolytes, sometimes include residues of opposite charges or neutral residues. Some polyelectrolytic IDRs have charge density of unity. Additional polyelectrolytes that are essential to function may include non-protein biopolymers. For example, inorganic polyphosphate [17] or matriglycans [18] are long negatively charged polyelectrolytic polymers composed of phosphates and saccharide building blocks and are involved in various distinctive functions.

The polyelectrolytic IDRs of proteins are positively charged when comprising repeating Lys (K) or Arg (R) residues (K/R repeats), whereas they are negatively charged when comprising repeating Asp (D) or Glu (E) residues (D/E repeats). A recent study [16] showed that many proteins include such repeats and that D/E repeats are more common than K/R. In eukaryotic genomes, ~10% of proteins have D/E repeats containing at least five residues; however, only ~5% of K/R repeats are at least five-residue long. D/E repeats are even more favored in longer polyelectrolytic IDRs. In various eukaryotes, about 1–2% of proteins include D/E repeats longer than 10 residues, but the population of K/R repeats containing 10 or more residues is zero (see Figure 1) [16]. Several proteins include 40–50-residue D/E repeats, but K/R repeats longer than 10 residues are not found in any organism. Several possible explanations have been proposed for why negatively charged D/E repeats are longer and more common than positively charged K/R repeats, including suggestions that K/R repeats are more prone to proteolysis [19] and that they may slow down translation kinetics in the ribosome because its exit tunnel is negatively charged [20,21].

The strong preference for D/E repeats over the K/R repeats is accompanied by a preference for E over D. In the human proteome, the frequency ratio of n(E)/n(D) is 3.1 in D/E repeats longer than 10 residues, whereas the overall ratio for human proteins of any length is 1.5. Similar values were found for the mouse proteome [16]. In K/R repeats, the n(K)/n(R) ratio is 1.7, and it is ~1 in all human proteins. The strong preference for D/E repeats over K/R and for E over D is supported but cannot be fully explained by the total concentrations of these amino acids as free solvated molecules in the cell (the concentrations of E, D, R, and K are 96 nM, 4.2 nM, 0.57 nM, and 0.4 nM, respectively) [22].

To elucidate the observed differences in the abundance and length of D/E and K/R repeats as well as the greater abundance of E in these repeats, here, we examined conformational ensembles of polyelectrolytic homo-polypeptides comprising D, E, K, or R. Using atomistic molecular dynamics (MD) simulations, we investigated the molecular biophysics of these homo-polypeptides to address whether the observed differences in D/E and K/R repeats may have a biophysical origin.

## 2. Materials and Methods

### All-Atom Molecular Dynamics Simulations

To quantify the biophysical properties of polyelectrolytes, we constructed polypeptides of length of 30 amino acids that were homo-repeats of aspartate (polyD), glutamate (polyE), arginine (polyR), or lysine (polyK). As a control, we constructed a polypeptide with consecutive repeats of glycine and serine, termed polyGS. The polypeptides were initially modeled as linear chains in PyMol, with more realistic conformations achieved during the MD simulations.

The conformational dynamics of the polypeptides were studied using all-atom MD simulations. The simulations were performed using GROMACS [23] version 2022. The molecular system was solvated in a box with periodic boundary conditions containing pre-equilibrated TIP3P water molecules, as implemented in the Charmm36m force field. Three salt concentrations were investigated. The salt concentration referred to as 0 M represents a neutral system, which was obtained by modeling the polyelectrolytes in an environment that included sufficient Na^+^ or Cl^−^ counterions to neutralize the charges on the homo-polypeptide amino acid residues. The other two salt concentrations involved modeling the polyelectrolytes in a low-salt (125 mM NaCl or MgCl_2_) or high-salt (250 mM NaCl or MgCl_2_) environment. We used the Charmm36m [24] force field. The LINCS algorithm [25] was used to control bonds during the simulation. The leapfrog algorithm was employed with steps of 2 fs.

The temperature was controlled at 300 K using a modified scheme of the theBerendsen thermostat [26]. The system was minimized using the steepest descent algorithm. Next, the system was equilibrated under an NVT ensemble and an NPT ensemble (100 ps each phase). Production runs were executed at a constant pressure (1 atm) for 200 ns. We ran each system to obtain five repeats at three NaCl concentrations (0 mM, low, and high) and five further repeats at two MgCl_2_ concentrations (low and high) for an accumulated simulation time of 25 μs.

Data analysis was performed using in-house python scripts. Principal Component Analysis (PCA) was performed as implemented in MDAnalysis [27].

## 3. Results

### 3.1. D/E Repeats Are More Common Than K/R Repeats

The bioinformatic analysis of the human proteome revealed that there are more proteins with negatively charged IDRs (D/E repeats) than with positively charged IDRs (K/R repeats) [16]. Figure 1 shows the number of proteins containing D/E or K/R repeats of various lengths (L_DE/KR_). Each data point in Figure 1 corresponds to all repeats with length ≤ L_DE/KR_. The shortest repeat considered in this analysis is of length 10. Figure 1 shows that for a length threshold of 10 consecutive residues, there are >250 proteins with D/E repeats but only ~10 proteins with K/R repeats. For all repeat lengths, a greater number of IDRs contain D/E repeats compared with K/R repeats. Similar results were reported for 22 different proteomes [16].

### 3.2. Dimensions of Polyelectrolytic Polypeptides

Guided by the observation that D/E repeats are often longer than K/R repeats, we explored the possibility that the preference for negatively charged polyelectrolytes over positively charged polyelectrolytes has a biophysical origin. For that purpose, we constructed 30-residue models of homo-polypeptides of polyelectrolytes containing either negatively (i.e., polyD and polyE) or positively (i.e., polyK and polyR) charged residues. The conformational ensemble of each of the homo-polypeptides was sampled using atomistic all-atom MD simulations that were analyzed to quantify their biophysical characteristics. As a control, we also simulated a polypeptide with 15 consecutive pairs of glycine and serine to produce a 30-residue polyGS.

Importantly, whereas the radius of gyration (Rg) of charged polypeptides was found to be in the range of 20–25 Å in the absence of salt and at both salt concentrations, the Rg of the polyGS control was found to be only ~10 Å. Thus, it appears that the more-extended dimensions of polyD/E and polyK/R are due to their polyelectrolytic nature. Moreover, the simulated ensembles of the polyelectrolytic polypeptides reveal differences between them. With respect to the negatively charged polypeptides, the Rg of polyE is larger than that of polyD (Rg_polyE_ > Rg_polyD_). For the positively charged polypeptides, polyK is more expanded than polyR (Rg_polyK_ > Rg_polyR_) (Figure 2A). Electrostatic repulsions between the charged amino acid residues of the polyelectrolytic polypeptides provide a possible physical explanation for the greater expansion of the polyelectrolytic polypeptides compared with the uncharged control (Figure 2A), whereas the screening of these repulsions by salt may explain the decrease in the value of Rg with the increase in the concentrations of NaCl from 0 to 0.25 M (Figure 2B).

Although the Rg analysis illustrates a clear difference between negatively charged polyD and polyE and an even greater difference between positively charged polyR and polyK, there is no clear difference between negatively and positively charged polyelectrolytes.

### 3.3. Conformational Ensemble of Polyelectrolytic Polypeptides

To further quantify the differences between different polyelectrolytic polypeptides, we performed PCA to elucidate the conformational ensemble of each system. Figure 3 shows the projection on the first two PCs of polyD (top row, orange circles) and polyR (bottom row, cyan circles) with no salt (left panels) and at a high salt concentration (right panels). As a control, we show in the background of each panel (gray circles) the projection on the first two PCs for the corresponding polyGS system. The PCA shows that the conformational ensembles of polyD and polyR are more restricted in low salt concentrations than high salt concentrations because of the greater screening of electrostatic repulsions in the presence of salt that allows a larger conformational space to be sampled with both more compact conformations than those sampled at low salt concentrations. The polyGS control samples a larger conformational space, which can be understood based on the absence of inter-residue electrostatic repulsions, thus a more flexible conformational ensemble. For the polyelectrolytes, the conformational ensemble is more restricted, likely due to electrostatic repulsions. The compaction observed upon the increase in salt concentration is illustrated to the right of each PCA by the presentation of a selected conformation for each system.

### 3.4. Flory Exponents and Relaxation Times

In addition to the conformational properties of the polyelectrolytic homo-polypeptides, their polymeric properties may also depend on their chemical nature. According to Flory [28], the Rg of a polymer scales with the number of bonds in the polymer (N) and an exponent ν$,\;Rg\propto N^{\nu }$. Due to the fractal nature of proteins in a good solvent, a similar relation can be obtained by calculating Rg as a function of the inter-residue distance in a single chain [29]. Hence, we use $Rg\propto {\left|i-j\right|}^{\nu }$, where |i−j| is the sequence separation between two residues in the substituent chain. Hence, by plotting Rg against |i−j| on a log–log scale, the Flory exponent can be derived from the slope (Figure 4A, right panel). Polymer theory predicts a scaling of ν = 1/3 for a compact polymer, ν = 2/3 for a random coil polymer, and ν = 1 for an extended conformation.

We used this relation to derive the Flory exponent for the simulated polyelectrolytes at three different salt concentrations (Figure 4A). With no salt and at both salt concentrations, the value of ν for the charged polypeptides lies in the range of 0.8–0.9, which is very similar to the value expected for a polyelectrolyte in an extended conformation because of extensive inter-residue charge repulsions. By contrast, the value of ν for polyGS is ~0.5, which is similar to the value expected for a random coil polymer. The Flory exponents are smaller for polyD and polyR than for polyE and polyK, in agreement with their Rg behavior (Figure 2). The Flory exponent decreases at higher salt concentration for all polyelectrolytes, but the effect is the largest for polyR.

The differences among polyD, polyE, polyR, and polyK were also demonstrated when quantifying polypeptide dynamics by analyzing the relaxation times, τ, of Rg (Figure 4B), calculated by fitting the auto-correlation function of Rg to a single exponential function (Figure 4B, right). Higher relaxation times are indicative of slower conformation sampling. The relaxation times increase with salt concentration, which can be rationalized by reducing the electrostatic repulsion among the charged homo-polypeptides. Figure 4B shows that the relaxation times are higher for polyD and polyR than for polyE and polyK, with polyGS having the largest τ value irrespective of salt concentration.

### 3.5. Sensitivity to Cation Valency Is Greater for D/E Repeats Than for K/R Repeats

An important question remains as to whether there is a direct connection between the salt concentration and the biophysics of the polyelectrolytes. To address this question, we plotted the mean Rg of each polyelectrolyte as a function of the number of ions adsorbed on the polypeptide (Figure 5A). Each point in Figure 5A was obtained using simulations at different salt concentrations, increasing from left to right. For polyD and polyE, the x-axis shows the number of Na^+^ (filled circles) or Mg^2+^ (empty circles) ions, and for polyK and polyR, the x-axis shows the number of Cl^−^ ions. Rg decreases as the number of adsorbed ions on the polypeptides increases, that is, the dimensions of the polypeptides decrease because the salt ions screen the electrostatic repulsions between neighboring amino acids. The positively charged polyelectrolytes (i.e., polyK and polyR) adsorb, on average, twice as many ions as their negatively charged counterparts (i.e., polyD and polyE), even though polyK is as compact as polyE. The greater compactness of polyD compared with polyE can be explained by the higher number of Na^+^ adsorbed on the former. However, the greater compaction of polyR compared with polyK cannot simply be explained by different extents of ion adsorption.

Thus far, we did not observe any significant difference between negatively and positively charged polyelectrolytes. However, a plot of Rg against the number of ions for polyelectrolytes in the presence of NaCl compared with MgCl_2_ shows that the Rg values of polyD and polyE decrease from ~22 Å in NaCl to 18 Å in MgCl_2_, whereas for polyK and polyR, the Rg values are less affected by changing the cations from monovalent Na^+^ to divalent Mg^2+^ (Figure 5A, filled vs. empty circles). We note that the adsorption of ions on uncharged peptides (i.e., polyGS) is negligible. The number of adsorbed mono- or divalent ions on polyGS ranges between 0 and 1 ions, regardless of the ionic strength.

Figure 5B shows a representative 2D distribution of polyE at 0.125 M NaCl and 0.125 M MgCl_2_, again showing the strong effect of cation valency on Rg for a negatively charged polyelectrolytic IDR.

## 4. Discussion and Conclusions

In this study, we investigated the conformational and polymeric properties of two positively charged homo-polypeptides (polyK and polyR) and two negatively charged homo-polypeptides (polyD and polyE). These charged homo-polypeptides are similar to polyelectrolytic sequences found in natural proteins, which often comprise repeats of K or R and of D or E. Some natural polyelectrolytic IDRs have high charge density per residue, but it is lower than unity, as they comprise neutral residues or residues with opposite charge. Here, we only focused on polyanionic and polycationic sequences, which are widespread in natural proteins. These stretches are often attached to folded domains and thus affect their function [30,31]. Quantifying the molecular biophysics of isolated polyelectrolytic peptides is essential towards understanding their role in biomolecular function, for example, via intra- or inter-molecular binding to other domains, either folded or disordered.

The current computational characterization of polyK, polyR, polyD, and polyE was motivated by a recent bioinformatic study that showed substantial differences between D/E and K/R repeats. K/R repeats were found to be much shorter and less common than D/E repeats. Although several potential biological explanations have been suggested to address these differences, here, we quantify their conformational properties to examine the possibility that the bias towards D/E repeats has a molecular biophysical basis.

Atomistic MD simulations show that the conformations adopted by the four charged homo-polypeptides are extended compared with typical neutral IDP conformations. This is illustrated by their respective mean Rg values, which are at least two times greater for polyelectrolytic peptides than for polyGS. The extended conformations are also reflected in the Flory exponent values of ~0.9 for polyelectrolytes compared with ~0.5 for polyGS. The difference between polyelectrolytic peptides and uncharged IDPs originates, as expected, from intra-molecular electrostatic repulsions, which also lead to a smaller conformational space. This electrostatic repulsion can be modulated by increasing the salt concentration. Increasing the concentration of NaCl results in the polyelectrolytic peptides adopting more compact conformations, with a lower Flory exponent, as well as in greater conformational heterogeneity.

Our study reveals some differences between the two positively charged homo-polypeptides and between the two negatively charged homo-polypeptides. Within the positively charged pair, polyR is more compact than polyK, whereas within the negatively charged pair, polyD is more compact than polyE. In addition, polyR is more sensitive to salt concentrations than polyK. This greater response to salt is also found for polyD compared with polyE, but to a lesser extent. The effect of salt on polyR and polyD correlates with the higher tendency of these polyelectrolytes to adsorb ions (Na^+^ and Cl^-^ by polyD and polyR, respectively).

Furthermore, a clear difference between the positively (polyK and polyR) and negatively (polyD and polyE) charged homo-polypeptides is observed when the simulation involves a divalent cation (Mg^+2^). Although all homo-polypeptides adsorb a similar number of ions when simulated in the presence of MgCl_2_, the negatively charged homo-polypeptides become much more compact compared with the effect observed when simulated with NaCl. Recently, a computational study of the solvation of isolated D, E, K, and R reported a more favorable hydration free energy for D and E than for K and R [32]. Furthermore, the heat capacities of the hydration of D and E have an opposite sign to those of K and R. The negative heat capacities of D and E have been attributed to differences in the hydration structure and the propagation of these effects beyond the first hydration shell. Our study also shows a higher tendency of D to adsorb both monovalent and divalent cations than E. This is in accordance with a recent study showing a greater number of calcium ions next to D than next to E, which was argued to explain their different roles in biomineralization processes [33].

In summary, alongside biological explanations for the abundance of D/E repeats over K/R repeats as possibly arising from their providing greater resistance to proteolysis or enabling more efficient translation by the ribosome [16], the current study also identifies biophysical differences between them. D/E repeats may have a more favorable solvation energy but are also more sensitive to cation valency and its effects on their degree of compaction. The abundance of polyelectrolytic peptides in various proteins may suggest that the understanding of their functional role is incomplete. The function and biophysical characteristics of polyelectrolytic peptides should be further addressed in the future both for polyelectrolytic homo- and hetero-peptides. The effect of the composition and pattern of Asp and Glu in polyelectrolytic hetero-peptides (or of Arg and Lys in polyelectrolytic hetero-peptides) on the biological function of polyelectrolytic peptides is unclear and may correspond to their specificity.

## Figures and Tables

**Figure 1 biomolecules-13-00363-f001:**
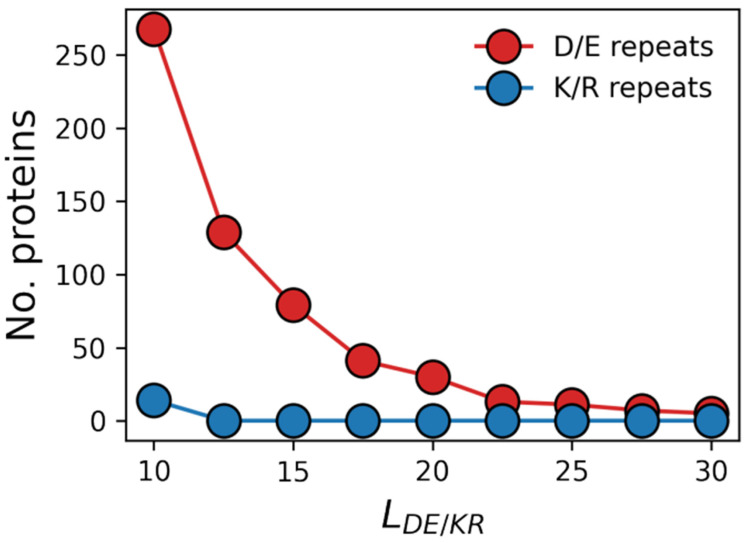
Occurrence of proteins with negatively or positively charged polyelectrolytic intrinsically disordered regions (IDRs) in the human proteome. Protein abundance is shown for proteins with D/E or K/R repeats of various lengths, as represented by L_DE/KR_ (i.e., the number of charged residues in the negatively or positively charged homo-polypeptides). The indicated number of proteins (out of the 20,600 proteins in the human proteome) is a cumulative value for all D/E or K/R repeat lengths up to the value of the corresponding L_DE/KR_. The shortest repeat length in this analysis is a repeat of 10 residues.

**Figure 2 biomolecules-13-00363-f002:**
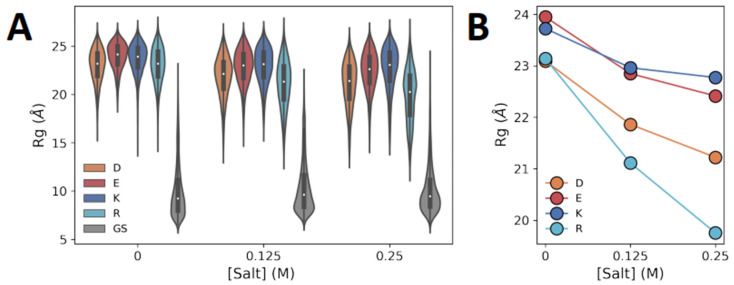
Dimensions of charged homo-polypeptides. (**A**) Violin plots of the Rg values of polyD, polyE, polyK, and polyR polyelectrolytic polypeptides, each constituting 30 residues, at three NaCl concentrations: 0 M, 0.125 M, and 0.25 M. The simulations at 0 M salt concentration included counterions to neutralize the charges of the homo-polypeptides. A polypeptide with 15 GS repeats was also simulated, as a control. The violin plots are colored according to amino acid identity, as indicated by the key. (**B**) Mean Rg of each charged homo-polypeptide as a function of NaCl concentration.

**Figure 3 biomolecules-13-00363-f003:**
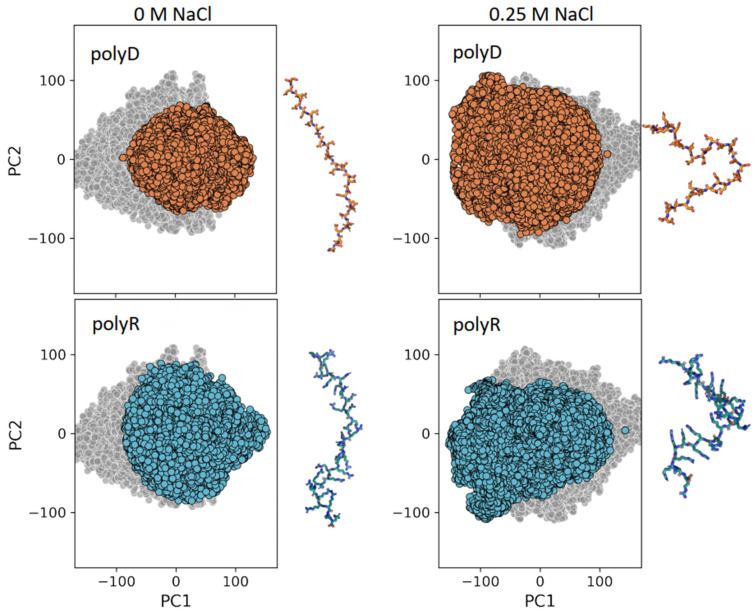
Conformational ensemble of charged homo-polypeptides. Projection of the first two principal components (PCs) from principal component analysis (PCA) of polyD (orange) and polyR (cyan) at NaCl concentrations of 0 M (**left**) and of 0.25 M (**right**). The projection for polyGS (gray) at the corresponding salt concentration is shown in the background of each panel for reference. Adjacent to each panel, a representative conformation is shown for each polyelectrolyte.

**Figure 4 biomolecules-13-00363-f004:**
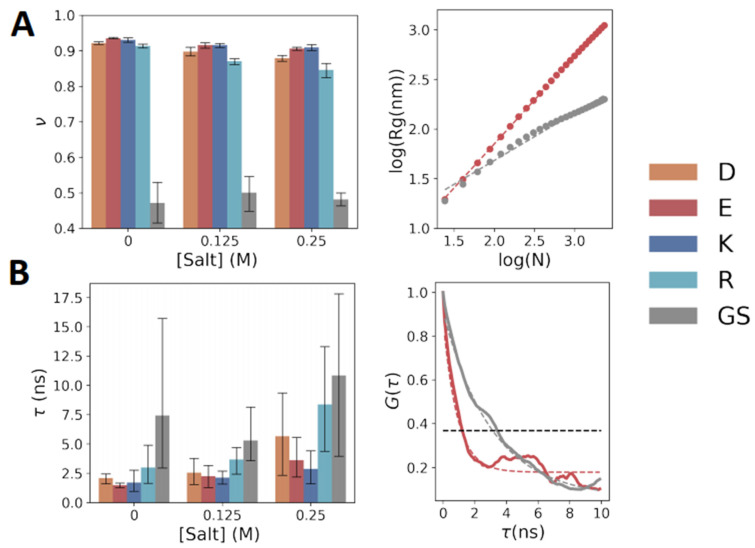
Polymeric properties of charged homo-polypeptides. (**A**) (Left) Flory exponent, υ, of the five simulated polypeptides at three salt concentrations, extracted from the relation Rg ~ |i-j|^υ^ (see main text for details). Error bars are the standard deviation of υ obtained from five independent simulations for each polypeptide. (Right) Representative example of the extraction of υ from the slope when plotting Rg versus |i-j| on a log–log plot. Data are shown for polyE (red circles) and polyGS (gray circles), and the dashed line is the best linear fit. (**B**) Relaxation times for Rg at three different salt concentrations. Values of τ were extracted by fitting the auto-correlation function, G(t), of Rg to a single exponential function (example on right panel for polyE and polyGS).

**Figure 5 biomolecules-13-00363-f005:**
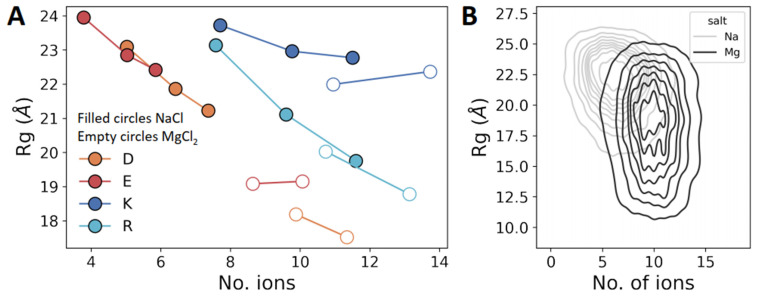
Ion adsorption on charged homo-polypeptides. (**A**) Mean Rg for each system as a function of the mean number of ions adsorbed on each charged homo-polypeptide. The three data points for each charged homo-polypeptide were obtained using simulations at three different concentrations of NaCl (0 M, 0.125 M, and 0.25 M) and two salt concentrations for MgCl_2_ (0.125 M and 0.25 M). The highest number of adsorbed ions for each system corresponds to simulations at a salt concentration of 0.25 M, with the lowest number of adsorbed ions being found at a salt concentration of 0 M. Filled and empty circles correspond to NaCl and MgCl_2_, respectively. (**B**) Two-dimensional distribution of Rg versus number of adsorbed sodium (blue) or magnesium (orange) ions for polyE when simulated in the presence of 0.125 M NaCl or MgCl_2_. Ion adsorption is defined based on a cutoff distance of 4 Å of the ions from any peptide atom, and the number of adsorbed ions is quantified by averaging the ions that satisfy the cutoff throughout the analyzed trajectory.

## Data Availability

Not applicable.

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
