# Peer review of "Conformational Analysis of Charged Homo-Polypeptides"

_biomolecules, 2023, doi:10.3390/biom13020363_

Round 1

Reviewer 1 Report

The authors have analysed the conformations of charged homo-polypeptides using molecular dynamics simulations to see if the different prevalence and lengths of the amino acid repeats has a biophysical meaning. The idea for this nice analysis came from their previous bioinformatics study. They used different approaches to analyse the MD trajectories from the extensive sampling of the peptides in different salt concentrations, including parameters used for analyzing polymers. All in all, a well written, compact paper. I recommend a minor revision according to the comments that I have marked in the article pdf file attached.

Author Response

We are grateful for the reviewer's comments. We have followed all the comments and typos pointed out by the reviewer and addressed them in the revised manuscript.

Reviewer 2 Report

In this manuscript, the authors have performed molecular dynamics (MD) simulations with polypeptides (polyK, polyR, polyD and polyE) which are found to be present in the disordered regions of many proteins. They have shown that salt concentration in the medium has a significant impact on conformational preference and dynamics of polypeptides, especially polyD and polyE. This is most likely due to the adsorption of divalent metal cations.

The manuscript is well-written, and the addressed problem is very fundamental. But I do not find the work sufficient to be published in “biomolecules”. I would suggest to accept the manuscript after major revisions. 

Major comments

1.     In the disordered region of the human proteome, both polyampholytic and polyelectrolytic residues are present. In this manuscript, the authors have shown results only for polyelectrolytic residues. The authors should design some polyampholytic peptides and compare them with current results.

2.     In the actual protein, IDR is attached to a structured part. Does it have any effect on a conformational ensemble of polypeptides?

3.     In the live cells, the environment is more crowded than the buffer condition used in the simulations. Does the crowding environment affect the conformational preference of the polypeptides?

Minor comments

1.     How do the authors define that the ion gets adsorbed? What is the residence time of an ion to a residue?

2.     How does the Rg vs No. of ions look like for control peptide, polyGS?

3.     In figure-5 legend, three different conc. has been mentioned for MgCl2, but in the figure, only two points (two empty circles) have been shown.  

Author Response

Following is our response to the reviewer's comments:

  1. In the disordered region of the human proteome, both polyampholytic and polyelectrolytic residues are present. In this manuscript, the authors have shown results only for polyelectrolytic residues. The authors should design some polyampholytic peptides and compare them with current results.

Our study focuses on polyelectrolyets and therefore polyampholytes is beyond the scope of the current study as they are expected to be quite different. Nonetheless, we have comment about this difference between polyelectrolytes and polyampholytes. 

2. In the actual protein, IDR is attached to a structured part. Does it have any effect on a conformational ensemble of polypeptides?

This topic is beyond the scope of the current study and deserves and independent investigation. A comment was added to the discussion of that aspect.

3. In the live cells, the environment is more crowded than the buffer condition used in the simulations. Does the crowding environment affect the conformational preference of the polypeptides?

Crowding is interesting but beyond the scope of the current study that focuses on the differences between polyK, polyR, polyD and polyE. It is not intuitive that crowding will influence them differently. 

Minor comments

  1. How do the authors define that the ion gets adsorbed? What is the residence time of an ion to a residue?

The details were added to the figure caption on the definition of adsorbed ions.

  1. How does the Rg vs No. of ions look like for control peptide, polyGS?

We have added some details on the number of adsorbed ions to polyGS in the main text. 

  1. In figure-5 legend, three different conc. has been mentioned for MgCl2, but in the figure, only two points (two empty circles) have been shown.

Following the reviewer comment, we have revised the figure caption.

Round 2

Reviewer 2 Report

I would recommend accepting the manuscript as it is.